# Recognition of ATT Triplex and DNA:RNA Hybrid Structures by Benzothiazole Ligands

**DOI:** 10.3390/biom12030374

**Published:** 2022-02-27

**Authors:** Iva Zonjić, Lidija-Marija Tumir, Ivo Crnolatac, Filip Šupljika, Livio Racané, Sanja Tomić, Marijana Radić Stojković

**Affiliations:** 1Laboratory for Biomolecular Interactions and Spectroscopy, Division of Organic Chemistry and Biochemistry, Ruđer Bošković Institute, Bijenička cesta 54, 10000 Zagreb, Croatia; iva.zonjic@irb.hr (I.Z.); tumir@irb.hr (L.-M.T.); icrnolat@irb.hr (I.C.); 2Faculty of Food Technology and Biotechnology, University of Zagreb, Pierottijeva 6, 10000 Zagreb, Croatia; fsupljika@pbf.hr; 3Department of Applied Chemistry, Faculty of Textile Technology, University of Zagreb, Prilaz Baruna Filipovića 28a, 10000 Zagreb, Croatia; lracane@ttf.hr; 4Laboratory for Protein Biochemistry and Molecular Modelling, Division of Organic Chemistry and Biochemistry, Ruđer Bošković Institute, Bijenička cesta 54, 10000 Zagreb, Croatia; Sanja.Tomic@irb.hr

**Keywords:** benzothiazoles, competition dialysis, DNA:RNA hybrids, ATT triplex, circular dichroism spectroscopy, RNase H

## Abstract

Interactions of an array of nucleic acid structures with a small series of benzothiazole ligands (bis-benzothiazolyl-pyridines—group 1, 2-thienyl/2-benzothienyl-substituted 6-(2-imidazolinyl)benzothiazoles—group 2, and three 2-aryl/heteroaryl-substituted 6-(2-imidazolinyl)benzothiazoles—group 3) were screened by competition dialysis. Due to the involvement of DNA:RNA hybrids and triplex helices in many essential functions in cells, this study’s main aim is to detect benzothiazole-based moieties with selective binding or spectroscopic response to these nucleic structures compared to regular (non-hybrid) DNA and RNA duplexes and single-stranded forms. Complexes of nucleic acids and benzothiazoles, selected by this method, were characterized by UV/Vis, fluorescence and circular dichroism (CD) spectroscopy, isothermal titration calorimetry, and molecular modeling. Two compounds (**1** and **6**) from groups 1 and 2 demonstrated the highest affinities against 13 nucleic acid structures, while another compound (**5**) from group 2, despite lower affinities, yielded higher selectivity among studied compounds. Compound **1** significantly inhibited RNase H. Compound **6** could differentiate between B- (binding of **6** dimers inside minor groove) and A-type (intercalation) helices by an induced CD signal, while both **5** and **6** selectively stabilized ATT triplex in regard to AT duplex. Compound **3** induced strong condensation-like changes in CD spectra of AT-rich DNA sequences.

## 1. Introduction

Nucleic acids are molecular targets for many drugs in cancer therapy due to their essential functions in cells (replication, transcriptional and translational regulation, and enzymatic reactions) [1]. Nucleic acid structures present a wide variety of shapes with varying major and minor groove widths that can be recognized by small molecules using a non-specific (mainly electrostatic) binding along the nucleic acid exterior, a specific groove binding, and intercalation (insertion of planar aromatic molecules between base pairs) [2].

Many studies have been directed towards the rational design of small molecules that will selectively recognize multistranded structures of nucleic acids, such as triplexes [2,3,4,5,6,7].

In most cases, triplexes in solution contain conformational features that are intermediate between A- and B-form. The parallel- or pyrimidine-motif (Py) has a C- or T-rich third strand bound in a parallel orientation to the duplex homopurine strand, while the antiparallel- or purine-motif (Pu) has the opposite orientation and a primarily A- or G-rich third strand [8,9].

Specific ligands can stabilize triple helices through intercalation. For example, it has been demonstrated that ethidium bromide stabilizes polydA–2polydT (ATT) with T⋅A × T triplets, while benzopyridoindole derivatives can stabilize triple helices containing both T⋅A × T and C⋅G × C^+^ base triplets [3]. The targeting of triplexes has recently been the focus of the antigene strategy for gene regulation. [10] The ability to target specific genes to modulate their structure and/or function in the genome has far-reaching implications in biology, biotechnology, and medicine (Figure 1) [11,12,13,14,15]. 

DNA:RNA hybrids are formed as intermediate structures during many biologically important processes, such as DNA replication, transcription, and telomere replication and replication of HIV by reverse transcription (Figure 1) [16,17,18,19,20]. They can also form R-loops that have been detected in various organisms from bacteria to mammals and play crucial roles in regulating gene expression, DNA and histone modifications, immunoglobulin class switch recombination, DNA replication, and genome stability [21]. Small organic molecules with the ability to selectively inhibit DNA replication via Okazaki fragments, thus also blocking transcription, have a great potential in treating cancer because the replication is often accelerated in cancer cells (Figure 1). In addition, small molecules selective for hybrid duplexes have potential therapeutic applications as telomerase and RNaseH inhibitors [22,23,24,25].

There are few examples in the literature dedicated to the discovery of compounds that selectively bind to DNA:RNA hybrids [26,27,28]. Literature sources point to the existence of a small number of ligands with selective binding to DNA:RNA hybrids [26,27,29,30,31].

In thorough studies of Arya and Chaires, a common structural motif that preferentially binds to the hybrid structures was identified employing rapid screening assays, the competition dialysis, and thermal denaturation of mixtures [26,27,29,31,32,33]. Several compounds containing the common motif-planar aromatic ring system with a “bay” region, such as ethidium bromide, coralyne, aminoglycoside, propidium, thiazole orange and ellipticine, demonstrated preferential binding to hybrid duplexes, among other nucleic acid structures. 

As the benzothiazole structure also meets this criterion of the common motif, we have chosen for this study nine benzothiazole derivatives, synthesized by Racane et al., [34,35,36], which demonstrated high antiproliferative activity on a panel of cancer cell lines. 

The main aim of this study was the detection of benzothiazole structure/s with preferential binding or spectroscopic response to DNA:RNA hybrids and ATT triplex in regard to regular (non-hybrid) DNA and RNA duplexes and single-stranded forms. Further, the mode of binding of selected ligands to DNA and RNA structures was determined using spectroscopic and calorimetric methods and molecular modeling. In this way, the mechanism of their antiproliferative activity [34,35,36] can be additionally clarified.

## 2. Materials and Methods

### 2.1. Spectroscopy Measurements

The UV/Vis spectra were recorded on a Varian Cary 100 Bio spectrophotometer (Agilent, Santa Clara, CA, USA), CD spectra on JASCO J815 spectrophotometer (ABL&E Handels GmbH, Wien, Austria) and fluorescence spectra on a Varian Cary Eclipse spectrophotometer (Agilent, Santa Clara, CA, USA) at 25 °C using appropriate 1 cm path quartz cuvettes. Absolute quantum yields were determined using software implemented with the instrument by the Integrating sphere SC-30 of the Edinburgh FS5 spectrometer in the quartz cuvette of a 10 mm path length, to avoid the scattering of incident light at the liquid–air interface, and testing solutions with a 2 mL volume were used. For ITC titrations, MicroCal™ VP-ITC (MicroCal, Inc., Northampton, MA, USA) was used. 

Polynucleotides were purchased as noted: poly rA, poly rU, poly dA, poly dT, poly rA–poly rU, poly (dAdT)_2_, poly dA−poly dT, poly (dGdC)_2_, and calf thymus (ct)-DNA (Sigma-Aldrich, St. Louis, MI, USA), and they were dissolved in Na-cacodylate buffer, *I* = 0.05 mol dm^−3^, pH = 7. According to the manufacturer’s instructions, the average length of polynucleotides in base pairs is ≥500. The calf thymus (ct)-DNA was additionally sonicated and filtered through a 0.45 mm filter [37]. Oligonucleotides (26-mers) were purchased as noted: oligo dA and oligo dT (Integrated DNA Technologies, Coralville, IA, USA) and dissolved in Na-cacodylate buffer, I = 0.05 mol dm^−3^, pH = 7. The DNA:RNA hybrid structures, DNA and RNA triplex, were prepared by mixing the two or three constitutive strands in sodium cacodylate buffer (*I* = 0.05 mol dm^−3^, pH = 7) with the addition of NaCl (0.05 M) and 1 mM EDTA, heating to 90 °C for 15′, and slow cooling to 10 °C. RNase H from Escherichia coli H 560 pol A1 was obtained from Roche/Merck in 25 mM Tris-HCl, 50 mM KCl, 1 mM dithiothreitol, 0.1 mM EDTA, 50% glycerol (*v*/*v*), pH 8.0 (1.0 U μL^−1^) and used directly.

Polynucleotide concentration was determined spectroscopically as the concentration of phosphates [38]. Spectrophotometric titrations were performed at pH = 7 by adding portions of polynucleotide solution into the solution of the studied compound for fluorimetric experiments, and CD experiments were done by adding portions of the compound stock solution into the solution of the polynucleotide. Even though most of the spectroscopic titrations were performed in sodium cacodylate buffer, *I* = 0.05 mol dm^−3^, there are a few exceptions, such as a fluorimetric experiment of **6** with poly dA−poly rU (pH = 7, sodium cacodylate buffer, *I* = 0.2 mol dm^−3^, and 1 mM EDTA) and with ATT (pH = 7, sodium cacodylate buffer, *I* = 0.05 mol dm^−3^, and 1 mM EDTA). For CD experiments, the exceptions are all titrations with ATT triplex and poly rA−poly dT hybrid (pH = 7, sodium cacodylate buffer, *I* = 0.05 mol dm^−3^, and 1 mM EDTA) and titration with poly dA−poly rU (pH = 7, sodium cacodylate buffer, *I* = 0.2 mol dm^−3^, and 1 mM EDTA). Emission was collected in the range λ_em_ = 350–600 nm. Values for *K*_s_ were obtained by processing titration data using the Scatchard [39] equation, all have satisfactory correlation coefficients (>0.99). Thermal melting curves for DNA, RNA, the 4 polynucleotide mix (from melting of mixtures experiment), and their complexes with studied compounds were determined as previously described by following the absorption change at 260 nm as a function of temperature [30]. The absorbance of the ligands was subtracted from every curve and the absorbance scale was normalized. *T*_m_ values are the midpoints of the transition curves determined from the maximum of the first derivative and checked graphically by the tangent method. The Δ*T*_m_ values were calculated by subtracting the *T*_m_ of the free nucleic acid from the *T*_m_ of the complex. Every Δ*T*_m_ value here reported was the average of at least two measurements. The error in Δ*T*_m_ is ±0.5 °C. 

For competition dialysis assay 13 different nucleic acid structures are used; each is placed in a separate Slide-A-Lyzer^®^ MINI dialysis unit, then in a flotation dialysis unit (Pierce Chemical Company, Dallas, TX, USA), and finally into a glass container, where it was dialyzed against a common ligand solution (concentration of a compound in solution is 5 μM) for 24 h at 25 °C. [40] When dialysis equilibrium is reached, the samples are pipetted from the dialysis unit to 96-well plate reader (Greinder) and SDS is directly added to dissociate bounds. Concentrations are determined by fluorescence on the Tecan microplate reader and visualized in Origin. 

RNase H assay was performed on a Varian Cary 100 Bio spectrophotometer at 37 °C using 1 cm path quartz cuvettes for 1–3 h. The reaction buffer was 50 mM Tris pH 8.0, 50 mM NaCl, and 10 mM MgCl_2_. Portions of compounds were added into the solution of the polynucleotide in glass cuvettes to achieve the desired experimental concentrations, then equilibrated at 0–4 °C for at least 12 h before use. After that, the polynucleotide (or compound-polynucleotide) solution was equilibrated at 37 °C; 2 μL of Rnase H was added directly and the reaction mixture mixed by gentle inversion 12 times. The sample was placed in the spectrophotometer block at 37 °C and A_260_ monitored for 1–3 h following an initial 2 min delay.

### 2.2. ITC Measurements

ITC experiments were performed using an isothermal titration microcalorimeter Microcal VP-ITC (MicroCal, Inc., Northampton, MA, USA) at 25.0 °C. Origin 7.0 software, supplied by the manufacturer, was used for data acquisition and treatment. In the titration experiments, aliquots of the ligand (see Appendix A for detailed protocol) were injected from a rotating syringe (220 rpm) into the calorimeter reaction cell containing the polynucleotide solution. The spacing between each injection was 600 s and the initial delay before the first injection was 2000 s. Blank experiments were carried out to determine the heats of the dilution of the ligands and the polynucleotides. All solutions used in the ITC experiments were degassed under vacuum before use to eliminate air bubbles. Each injection generated a heat burst curve (P in µW versus time). The data were imported to Origin 7.0 and the area under each peak was determined by integration to evaluate the heat associated with the injection. The data were corrected for heats of dilution. The resulting data were analyzed by using the Origin 7.0 software according to the model based on a single set or two sets of identical binding sites to estimate the binding constants (*K*_a_), the binding stoichiometry (N), and the enthalpy of binding (∆_r_*H*°). The reaction Gibbs energies (∆_r_*G*°) were calculated by using the following equation: ∆_r_*G*° = −*RT*ln(*K*_a_). Entropic contribution to the binding Gibbs energy was calculated by the equation: *T*∆_r_*S*° = ∆_r_*H*° − ∆_r_*G*°.

### 2.3. Molecular Modeling

ATT molecule was built using the structure of the triplex available in PDB with PDB_id 1d3x wherein, for the purpose of modeling, CYT and GUA were replaced by THY and ADE. The obtained molecule was parametrized within OL15 and BSC1 [41] force fields and minimized. ATT-ligand complexes were built in program Pymol [42]. The complexes were placed in the center of the octahedral box filled with TIP3P type water molecules [43]. A water buffer of 11 Å was used and Na^+^ ions were added to neutralize the systems. The solvated complexes were geometry optimized in 3 cycles using the steepest descent and conjugate gradient methods. After optimization, systems were equilibrated for 0.5 ns in two steps: during the first step of 50 ps, the system was heated from 0 to 300 K under NVT conditions. In the next step, the water density was adjusted (NPT conditions). The equilibrated systems were subjected to the 200 ns of productive, unconstrained MD simulations at a constant temperature (300 K) and pressure (1 atm) with the time step of 2 fs (SHAKE algorithm was used to restrain the motion of hydrogens). The simulations were performed with the pmemd program, available within the AMBER16 package, using periodic boundary conditions, wherein the electrostatic interactions were calculated using the particle-mesh Ewald (PME) method [44]. The temperature and pressure was regulated using Langevin thermostat [45] (with collision frequency of 1 ps^−1^) and the Berendsen barostat [46], respectively.

## 3. Results and Discussion

### 3.1. Characterization of Compounds in Aqueous Medium

This study included nine cationic compounds: bis-benzothiazolyl-pyridines (**1**–**3**, group 1), 2-thienyl/2-benzothienyl-substituted 6-(2-imidazolinyl)benzothiazoles (**4**–**6**, group 2), and 2-aryl/heteroaryl-substituted 6-(2-imidazolinyl)benzothiazoles (**7**–**9**, group 3) (Figure 1). All compounds were soluble (*c* = 5 × 10^−3^ mol dm^−3^) in redistilled water or aqueous buffer (sodium cacodylate/HCl buffer, *I* = 0.05 mol dm^−3^). Buffered solutions of studied compounds were stable for more days. The absorbancies of studied compounds (Appendix A) were proportional to their concentrations up to *c =* 2 × 10^−5^ mol dm^−3^. Linear changes in absorption with the increase of concentration indicate that studied compounds do not aggregate by intermolecular stacking at the experimental conditions used. Emission, absorption maxima, and the corresponding molar extinction coefficients (ε) of all studied compounds are summarized in Table 1. The excitation spectra correspond to compounds’ absorption spectra in the area where emission and excitation spectrum do not overlap (Appendix A).

### 3.2. Study of Interactions of Benzothiazoles with Nucleic Acids in Aqueous Medium

#### 3.2.1. Competition Dialysis Assay with **1**–**9**

Competition dialysis assay is a potent and valuable quantitative tool for examining compounds of interest that recognize selective structure/sequence of nucleic acid. In this method, an array of nucleic acid sequences and structures is used. Each is placed in a separate MINI dialysis unit fixed in a flotation dialysis supporter inside a glass container and dialyzed against a ligand solution. The free ligand solution is the same for all the structures, but when equilibrium is reached, each of the structure will bind the ligand according to its binding affinity [29,40]. 

Nine previously synthesized [34,35,36] benzothiazole compounds with structural changes (Figure 2) in position 2 of the imidazole-based benzothiazole core were used to determine the effect of these modifications on the ability of compounds to interact with different DNA and RNA structures. 

However, the interaction of eight compounds with 13 different nucleic acid structures was studied (Figure 2): single-stranded and double-stranded polynucleotides, DNA:RNA hybrids, and DNA and RNA triplexes. The interactions of compound **9** with polynucleotides were not further characterized, as compound **9** aggregated in the aqueous solution during the experiment. 

The interaction of compounds with double-stranded polynucleotides depended on the base composition and secondary structure. Mostly, compounds demonstrated strong binding to DNA triplex ATT. Especially for **5**, and even more so, **6**, where binding with ATT triplex was stronger than with the double-stranded DNA, RNA, and DNA:RNA hybrids. As the amount of ligand bound to each nucleic acid structure (C_bound_) is directly proportional to the ligand binding affinity, it is clear that **1** and **6** displayed the highest affinities toward the majority of the nucleic acid structures. Regarding DNA:RNA hybrids, **6** demonstrated the strongest binding to poly dA−poly rU, while **1** bound slightly better to poly rA−poly dT than to poly dA−poly rU. Only **1** demonstrated preferential binding to single-stranded poly dT, while both **1** and **6** exhibited stronger binding to poly rA. 

Two metrics were used, the specificity sum, SS and the ratio C_max_/SS, to gain information about the structural selectivity and compound affinity. To calculate the specificity sum, the binding data first need to be normalized relative to the maximal amount bound (C_max_) to any of the structures in this assay. Then normalized amounts for each nucleic acid structure in the assay were simply summed. As 13 nucleic acid structures were used in this experiment, the SS ratio can range from 1, which denotes the binding to only one nucleic acid structure, to 13, which means an equal binding to all structures. According to Figure 3, the best structural selectivity demonstrated **5** and **6**. Values for the specificity sum for all studied compounds are shown in Figure 3.

Ratio C_max_/SS refers to both affinity and selectivity. This ratio is directly proportional to binding affinity. Thus, if C_max_ is large (high binding affinity) and SS is small (high selectivity), a high value of C_max_/SS will be obtained and vice versa [29,40]. 

Identification of compounds with the best-combined selectivity and affinity can be obtained by comparison of SS and C_max_/SS values. Based on these metrics, **5** and **6** were identified as compounds with the greatest combined selectivity and affinity. Despite the largest SS value, **1** was also selected for further characterization with polynucleotides, as it demonstrated relatively higher C_bound_ values (amount of ligand bound to each nucleic acid structure), determined for ds- and triplex polynucleotides than other compounds. All three compounds demonstrated selectivity for ATT triplex. Additionally, **6** exhibited the highest affinity toward poly dA−poly rU, while **1** displayed a higher affinity toward poly rA–poly dT.

#### 3.2.2. Fluorescence Spectroscopy and Isothermal Titration Calorimetry

Isothermal titration calorimetry and fluorescence spectroscopy have been used to characterize the ligand binding to nucleic acid structures [48,49]. According to the results of the competition dialysis assay (affinities and selectivities), several complexes were selected for detailed characterization: **1**-ATT (Figure 4), **1**-poly rA−poly dT, **6**-ATT, **6**-poly dA−poly rU, and **5**-ATT (Table 2).

Binding affinities of **5** and **6** could not be calculated by ITC titrations due to the aggregation of these compounds in the ITC method concentration range that interfered with obtaining reliable results. For **6**, binding affinities were assessed by fluorescence spectroscopy, which enabled measurements in lower sample concentrations. The fluorescence changes of **5** in titration with ATT triplex were too small for the accurate calculation of the binding constant.

ITC titration experiments resulted in mostly negative peaks, indicating that the binding processes were exothermic (Figure 4). The resulting data for the **1**-poly rA−poly dT complex was fitted to a single-site binding model, while data for **1**-ATT was fitted to model two sets of sites by using a nonlinear least square method (Table 2). The analysis of ITC experiments of compound **1** with poly rA−poly dT and ATT triplex demonstrated high and similar binding affinities (log *K**s*, Table 2), which is consistent with the results from competitive dialysis assay.

Compound **1** demonstrated two types of binding in titration with ATT triplex (Figure 4). The first binding event was characterized by a higher binding constant than the second type of binding. The first binding event is an entropically guided process probably accompanied by the release of bound water molecules from the polynucleotide groove to the bulk [48]. The second type of binding, characterized by a higher ratio N, is an enthalpy-driven process accompanied by an increase in the number of hydrogen bonds, aromatic stacking, electrostatic interactions, and van der Waals interactions, followed by a large favorable entropy contribution. The interaction of **1** with poly rA−poly dT was, similar to its complex with ATT, characterized by positive (favorable) binding entropy and weak negative enthalpy (Table 2), indicating an entropically driven process. 

The groove binding is usually entropically favorable and slightly endothermic, resulting from the release of relatively highly ordered water molecules surrounding the apolar surfaces to the bulk. The enthalpy contribution to the free energy is associated with the overall increase of bonding (hydrogen bonds, ionic, electrostatic and van der Waals interactions, and polarization of the interacting groups) [51]. 

An analysis of the fluorescence data of **6** with poly dA−poly rU and ATT polynucleotides gave high binding constants and rather high ratios, N, for both complexes. Unlike **6**, **5** induced small emission changes upon binding to ATT triplex, thus disabling calculation of the stability constant.

#### 3.2.3. Thermal Melting Experiments and RNase H Assay

Thermal denaturation is an additional method for monitoring binding to nucleic acids [52]. Stabilization of nucleic acid structure induced by small molecules causes an increase in the melting temperature of that structure. In a typical experiment, a ligand is monitored against one nucleic acid structure at conditions close to the equimolar ligand/nucleic acid ratio. The melting of mixtures, a relatively straightforward extension of the typical thermal melting experiment design, enables an evaluation of the stabilization effect of the ligand against a number of different nucleic acid structures. If there is an excess of nucleic acid over the compound where the binding sites are not fully saturated, a preference toward sequence or structure can be established [40]. 

In this experiment, ligand selectivity was studied with a mixture of four different polynucleotides, DNA (poly dA–poly dT), RNA (poly rA–poly rU), and two DNA:RNA hybrids (poly rA–poly dT and poly dA–poly rU) (Figure 5, Table 3).

The compounds **1** and **5** demonstrated the stabilization effect of all studied ds-polynucleotides (Table 3, Figure 5). In addition, these two compounds demonstrated a better stabilization effect of poly dA−poly rU than of poly rA−poly dT. Interestingly, among ds-nucleic structures, **6** almost exclusively stabilized poly dA−poly rU. It has been demonstrated that the stability of hybrid duplexes, composed of homopurine and homopyrimidine strands, depends on several factors, such as the percentage of deoxyribo(pyrimidine), i.e., d(Py) content in each of the strands, the oligomeric length and the percentage of (A)*_n_*:(T or U)*_n_* content [53]. Thus, for example, the hybrid duplex containing the DNA purine strand and the RNA pyrimidine strand d(Pu):r(Py), such as poly dA−poly rU, demonstrates much less thermal stability compared to the r(Pu):d(Py) composition, such as poly rA−poly dT. This could be a reason for the higher stabilization effect of dArU, in comparison to rAdT, induced by **1**, **5**, and **6**. Among ds-polynucleotides, **1** demonstrated the most significant stabilization effect with poly dA−poly rU and especially poly dA−poly dT (Table 3). It should be emphasized that **1** possesses a higher number of net positive charges than **5** and **6** (**1**, +2; **5**, +1; **6**, +1). However, only **6** demonstrated selective stabilization of the hybrid duplex, poly dA−poly rU, in a mixture of double-stranded polynucleotides. 

As all three ligands stabilized poly rA−poly dT and poly dA−poly rU, we performed a spectrophotometric RNase H assay to evaluate the capability of **1**, **5**, and **6** to inhibit RNase H [29]. The cleaving of the RNA strand from a DNA:RNA hybrid duplex was followed by an increase in A_260_ (Figure 6).

The best inhibition activity of the RNase H digestion of poly rA−poly dT demonstrated benzothiazole **1**. The data from a thermal melting of polynucleotide mixtures show a good correlation with the results of the RNase H assay. The RNase H assay was utilized here as an additional method to the thermal melting of polynucleotide mixtures. It provides information on which ligands can be biologically relevant as RNase H inhibitors and deserves detailed investigation regarding that.

The melting profile of free ATT in sodium cacodylate buffer (50 mM sodium cacodylate buffer, 50 mM NaCl, and 1 mM EDTA) demonstrated biphasic transition (Figure 7, Table 4).

The first transition was at *T*_m1_ = 22.6 °C and corresponds to dissociation of the third strand (poly dT with Hoogsteen base pairs from the major groove). The second transition was at *T*_m2_ = 70.7 °C and corresponds to dissociation of Watson–Crick base pairs of double-stranded poly dA-poly dT [54,55]. Albeit **1** stabilized both the ATT triplex and poly dA−poly dT duplex, the higher stabilization effect was demonstrated with the ATT triplex. Unlike **1**, **5** and **6** increased *T*_m1_, while *T*_m2_ did not change. Particularly interesting was compound **6**, which stabilized ATT significantly even at very low ratios, **r** (Figure 7). Data from this experiment for poly dA−poly dT (Table 4) duplex agrees very well with those for the same polynucleotide obtained in the melting of mixtures experiment (Table 3).

#### 3.2.4. Circular Dichroism (CD) Experiments

Based on competition dialysis results, we further proceeded with the characterization of selected complexes by CD spectroscopy. Electronic circular dichroism (ECD) is highly sensitive toward conformational changes in the helical structure of DNA and RNA and their complexes with small molecules [56,57]. In addition, an induced CD spectrum (ICD) that can arise from the interaction of achiral small molecules such as **1**, **5**, and **6** with nucleic acids, could be very informative of the binding modes. We monitored changes in CD spectra upon the interaction of **1**, **5**, and **6** with representatives of the B-helix family, ctDNA with a mixed base pair composition, and with poly dA−poly dT characterized by a much narrower and deeper minor groove in comparison to other common B-helices. As a model of A-helical structure, we used poly rA−poly rU (ds-RNA) characterized by a wide and shallow minor groove and deep and narrow major groove [58,59]. Interactions were also studied with two double-stranded DNA:RNA hybrids, poly dA−poly rU, and poly rA−poly dT and the DNA triple helix, ATT. While NMR, Raman, and X-ray fiber diffraction studies suggest B-like conformation for poly rA-poly dT in high humidity conditions, it seems that the global conformation of poly dA-poly rU is considerably more affected by the ribopyrimidine strand, resulting in an A-type helix closer in conformation to the A-form of RNA [60,61]. Further, NMR and IR data suggest that structural characteristics of triplexes resemble B-DNA much more than A-DNA [62,63,64]. 

The addition of **1**, **5**, and **6** to polynucleotides mainly caused a decrease of CD intensity of DNA and RNA polynucleotides at their maximal values (ctDNA at 275 nm, AT-DNA, AU-RNA, and DNA:RNA hybrids at 260 nm) (Figure 8); however, induced CD spectra of these compounds, which changed with increasing ratio, ***r***, were more informational regarding their modes of binding. 

At **r** ≤ 0.1, **6** caused either negligible or weak negative ICD signals (around 320 nm) with all polynucleotides (Appendix A). However, at **r** > 0.1, this compound differentiates between B- and A-type helices by the mode of binding (Figure 8). Negative ICD signals support intercalation to A-type helices, poly dA−poly rU and poly rA−poly rU, whereas an appearance of a bisignate CD signal implied the formation of **6** dimers most probably inside the minor groove of poly rA−poly dT, poly dA−poly dT, and ATT triplex. Interactions of **6** were additionally investigated with ATT 26mer by fluorimetric and CD spectroscopy to estimate the influence of the ATT polymer length on the binding strength. Processing of titration data gave ratio *n* and binding constant (log *K*s = 7.4, *n* = 0.7) comparable to that obtained with the ATT polynucleotide (Table 2). Regarding CD titration of **6** with ATT 26mer, a similar bisignate ICD signal, as in the CD titration with ATT polynucleotide, was noticed, suggesting that ATT polymer length did not influence the mode of binding (Appendix A) [65].

Unexpectedly, **6** did not bind in the form of dimers to ctDNA (B-helix), instead, changes (negative ICD signals, Figure 8) were indicative for intercalation, which can probably be related to the composition of ctDNA containing 42% of GC basepairs beside AT base pairs. 

Unlike **6**, **5** did not differentiate among polynucleotide conformations (Appendix A). Its addition caused a rise of bisignate ICD signals with all studied nucleic acid structures. Such an effect supports a formation of dimers (minor groove of ATT triplex and major groove of poly rA-poly rU) or larger aggregates, similar to those observed with poly dA-poly dT and ctDNA.

On the other hand, bis-benzothiazolyl-pyridine **1**, which is sterically more demanding compared to **5** and **6**, provoked a strong positive ICD band (at 340 nm) at **r** ≤ 0.1, implying binding within the minor groove of ctDNA, poly dA−poly dT, and ATT triplex (Figure 8). At ratios higher than **r** = 0.1, an excess of **6** molecules cannot accommodate inside the minor groove so well; instead, **6** forms aggregate along the polynucleotide backbones. Negative ICD signals (around 340 nm) at **r** ≤ 0.1 point towards intercalative binding to poly rA−poly rU and both hybrids. An increase in ICD intensity with an increase of **r** (**r** > 0.1) suggests aggregation of **6** along the polynucleotide surfaces. Further, a clear isodichroic point (λ = 253 nm) observed for **1** and poly rA−poly dT strongly suggests one dominant interaction mode of this compound with the DNA chiral axis.

Due to the strong interaction of **1** with the ATT triplex, we decided to examine interactions of this triplex with its regioisomer **3** to see whether the position of benzothiazole-imidazolinyl chains on pyridine ring affected them. Interestingly, compound **3**, unlike its regioisomer **1**, induced a strong increase in the intensity of CD spectra of ATT triplex, poly dA−poly dT, and ctDNA (ctDNA at 275 nm, AT-DNA at 260 and 282 nm, and ATT at 282 nm) compared to the intrinsic CD bands of these polynucleotides, and in addition, strong positive ICD bands located around 350 nm (Figure 9, Appendix A). 

Similar strong changes in CD spectra of polynucleotides were noticed with bis-polyaza pyridinophane derivatives that, similarly to compound **3**, consist of two chains attached to pyridine as a central unit [66]. The reason that compound **3** induced condensation-like changes, while compound **1** did not, is probably the position of chains on the pyridine ring. Unlike **1**, **3** and bis-polyaza pyridinophane derivatives have two chains attached at the same positions on the pyridine ring (2 and 6 positions).

#### 3.2.5. Molecular Modeling

The binding of **6** dimers inside the minor groove of the ATT triplex was also examined by molecular modeling (Figure 10). The mode of binding of **6** with the ATT triple structure suggested by the spectroscopic methods was consistent with the results obtained by molecular modeling.

## 4. Conclusions

The search for small molecules with selective binding to specific sites in DNA or RNA structures is still of intense interest. Such binding can block or interfere with important processes, e.g., transcription, recombination, and DNA repair.

Competition dialysis assay [27,71,72] allows the straightforward evaluation of sequence and structural selectivity of different DNA and RNA binding ligands. In this study it enabled the detection of three benzothiazole compounds, (**1**, **5**, and **6**), with preferential binding to DNA:RNA hybrids and ATT triplex in regard to regular (non-hybrid) DNA and RNA duplexes and single-stranded forms. 

Compound **6** preferentially stabilized dArU hybrid among other ds-polynucleotides. RNase H assay confirmed the results of thermal melting experiment (Table 3) and identified ligand **1** as a potential RNase H inhibitor for the digestion of poly rA−poly dT.

While all three compounds demonstrated a strong stabilization effect of ATT triplex, only compound **6**, in both thermal melting experiments (Table 3 and Table 4), demonstrated selective binding to ATT triplex in regard to AT duplex. Such stabilization could be exploited to inhibit the gene expression involved in cancer, for interference with DNA replication or inducing transcriptional repression, site-specific mutations, and recombination [11].

To the best of our knowledge, among small molecules previously reported [73,74,75,76,77], **6** with the chlorobenzothiophene substituent has the largest stabilization effect on ATT triplex (Δ*T*_m_ = 44 at **r** = 0.1). In addition, **5** (with bithiophene substituent) and, especially, **6** selective stabilization of triplex in the form of dimer inside the minor groove, has not been reported yet. The mode of binding of **6** (inside the ATT minor groove in the form of dimer) was confirmed by CD spectroscopy and molecular modeling. These results were made with long polynucleotides (≥500 base pairs) that can provide a large excess of binding sites along the DNA helical structure and ease the determination of binding modes. To see if the same selectivity of molecule **6** toward ATT base triplets exists in shorter sequences, we performed measurements with the 26-nucleotide-long ATT triplex, possessing the same conformation (B-DNA, Figure 10) as its longer form. The fluorescence and CD experiments performed with 26 mer revealed high affinity and the same ICD profile as with longer sequences, confirming that the pattern of recognition is present regardless of nucleotide chain length.

As the AT-rich repeated sequences are often found in the sites for DNA replication initiation in bacterial, archaeal, and eukaryotic replicons, it would be interesting to confirm identified selectivity with a more distinct AT-rich oligomer sequence, for example, the GATCTATTTATTT of replication origin in *E. coli* [78]. However, this will be investigated in another study, after careful selection of the oligomers. Compound **6** could differentiate between B- and A-type helices by the mode of binding. While an appearance of negative ICD signals supported intercalation to A-type helices, poly dA−poly rU, and poly rA−poly rU, bisignate CD signal implied the binding of **6** in the form of dimers inside the minor groove of poly rA−poly dT, poly dA−poly dT, and ATT triplex.

In contrast to its regioisomer **1**, ligand **3** induced a strong increase in CD intensity of AT-rich sequences (ctDNA at 275 nm, AT-DNA at 260 and 282 nm, and ATT at 282 nm) and strong positive ICD bands around 350 nm (Appendix A). Similar condensation-like changes of the nucleic acid structure could be utilized in pharmaceutics for DNA delivery by viral or non-viral vectors in gene therapy [79,80,81]. Furthermore, DNA condensation can be applied in the construction of biosensors based on the liquid-crystalline properties of condensed DNA [82]. This result, as well as the identification of ligand **1** as a potential RNase H inhibitor, will be investigated in more detail and will be published elsewhere.

## Data Availability

Not applicable.

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
