# Peer review of "Recognition of ATT Triplex and DNA:RNA Hybrid Structures by Benzothiazole Ligands"

_biomolecules, 2022, doi:10.3390/biom12030374_

Round 1

Reviewer 1 Report

In an exploratory approach the authors investigated the binding properties of a selected series of benzothiazole derivatives with thirteen different nucleic acid forms. Some of these ligands were found to have special selectivity or affinity towards particular DNA/RNA forms, specifically towards triplex DNA/RNA. In general, the authors provide a large set of data from different complementary experimental and theoretical methods, that provide a good basis for the binding analysis (but see some special concerns listed below). As there is still the need for novel selective DNA/RNA binders that may give useful information for the development of nucleic acid-targeting drugs, this may be considered a relevant contribution to the field. Prior to publication, however, the following points have to be clarified in detail:

A series of compounds was investigated and they showed different binding behavior towards the tested nucleic acid structures. These results are certainly interesting, but they are meaningless unless a trend of structure-property relationships is deduced that results in clear and general design principles for DNA-RNA triplex binders.

Experimental data:

– Source and proof of purity of the employed ligands have to be provided.

– Competition dialysis: Fluorimetric analysis usually does not provide an accurate quantitative determination of ligand concentration/amount. Photometric analysis is certainly more reliable. Moreover, it should be described how mass balance was controlled, because the ligands can also adsorb to the dialysis membranes.

– Stokes shifts in nm are meaningless. They must be given in energy scale.

– Emission quantum yields are missing.

– Equations used for analysis purposes should be presented in the Supporting Information.

– Is Scatchard analysis really the best way to determine the binding constants? Please explain why no more sophisticated methods were used.

– Thermal melting analysis: The peak areas do not look like Gaussian functions. Are these raw data or were the peaks already approximated with a Gaussian peak function? Overall, some melting curves reveal only very weak effects in terms of absolute values, so I wonder whether these data can be used for quantitative analysis without error analysis.

– Figure S1: Some spectra have a significant break at ca. 360 nm. Please explain.

The choice of nucleic acids should be justified in more detail, especially as most of them are not really biologically relevant. Especially, after particular selectivities of ligands have been identified, more elaborate and distinct oligonucleotide sequences with the same DNA/RNA structure should have been used to substantiate the general statements about selectivity and affinity towards this nucleic acid form.

Figure 1: Although this figure is well organized and very informative, it is not necessary in this context and would rather serve its purpose in a review article.

The literature on triplex ligands is not complete; and the list of refs (2–7) also contains literature on G4 DNA ligands, that is incomplete as well.

It is very disturbing that Figures and Tables proceed over page breaks in the Supporting Information.

The conclusion is only a summary.

Author Response

Comment 1. Source and proof of purity of the employed ligands have to be provided.

Answer: Prof. dr. Livio Racané synthesized and characterized the compounds, and data on tested compounds (NMR data, ESI and elemental analysis) were published in three separate papers, references 30, 31 and 32 in previous version, 34, 35 and 36 in revised version. These references are listed in Introduction and Results and discussion (3.2. Study of interactions of benzothiazoles with nucleic acids in aqueous medium, 3.2.1. Competition dialysis assay with 1-9). Studied compounds are from the same synthetic batch as in those papers. Nevertheless, we have included the characterization of the tested compounds in SI to make it easier for readers to review this data in one place instead of in three different papers. The changes in SI were marked in yellow.

Comment 2.  Competition dialysis: Fluorimetric analysis usually does not provide an accurate quantitative determination of ligand concentration/amount. Photometric analysis is certainly more reliable. Moreover, it should be described how mass balance was controlled, because the ligands can also adsorb to the dialysis membranes.

Answer: We agree with referee that photometric analysis is generally more reliable than fluorimetric in term of quantitative determination of particular ligand concentration. However for competition dialysis method any analytical method could be used for concentration determinations, due to the properties of the ligand. This is because all structures and sequences are in equilibrium with the same ligand concentration after equilibration is established. Amounts of examined ligand bound to the particular structures are directly proportional to the particular ligand binding affinities. That is why competition dialysis gives quantitative measure of selectivity by distinguish the structure within the sequence array that is preferred by a particular ligand. Also, apparent binding constants and binding free energies can be quickly calculated from competition dialysis data. Yet, since they are derived from a single set of reactant concentration (obtained by any analytical method) these constants were validated by complete binding isotherms for rigorous insights. It was proved that complete binding isotherms were highly correlated with the apparent binding constants.

There is possibility that ligand binds to the surfaces (for example dialysis membranes), but minor adsorption can be tolerated, since it represents only another competing equilibrium. Only the free ligand concentration determines the amount bound to the various polynucleotide structures, so the assay remains valid as long as there is appreciable free ligand in solution. (J.B. Chaires, Competition Dialysis: An Assay to Measure the Structural Selectivity of Drug-Nucleic Acid Interactions, Current Medicinal Chemistry - Anti-Cancer Agents, 5 (2005) 339-352.; J.B. Chaires, A Competition Dialysis Assay for the Study of Structure-Selective Ligand Binding to Nucleic Acids, Current Protocols in Nucleic Acid Chemistry, 11 (2002) 8.3.1-8.3.8.)

Comment 3.  Stokes shifts in nm are meaningless. They must be given in energy scale.

Answer: Thank you for your suggestion. Stokes shifts which were in nm in Table 1 are given in energy scale.

Comment 4.  Emission quantum yields are missing.

Answer: Quantum yields are inserted in Table 1.

Comment 5.  Equations used for analysis purposes should be presented in the Supporting Information.

Answer: Equations are inserted in SI. The changes in SI were marked in yellow.

Comment 6.  Is Scatchard analysis really the best way to determine the binding constants? Please explain why no more sophisticated methods were used.

Answer: We do not claim that Scatchard analysis is the absolute best method for calculating binding constants, but it is often used to calculate binding constants of complexes, especially of small organic molecules and DNA and RNA molecules (Chaires JB. Analysis and interpretation of ligand-DNA binding isotherms. Methods Enzymol. 2001;340:3-22). Based on Scatchard's analysis, we can obtain not only the affinity of the receptor / substrate for the ligand but also the number of binding sites. Titrations are usually done in excess of polynucleotide binding sites to ligand concentration where it can be considered that ligands bind to isolated, mutually independent binding sites. Under these conditions, Scatchard's analysis gives reliable values of the binding constants and the number of binding sites, n ([bound compound]/[polynucleotide].

 Comment 7.  Thermal melting analysis: The peak areas do not look like Gaussian functions. Are these raw data or were the peaks already approximated with a Gaussian peak function? Overall, some melting curves reveal only very weak effects in terms of absolute values, so I wonder whether these data can be used for quantitative analysis without error analysis.

Answer: These are raw data and are not approximated with a Gaussian peak function. We simply determine the maximum of the first derivative of the absorbance signal (dA/dT) in Origin program which is sufficient for a qualitative analysis of the data, meaning an increase in thermal melting temperature indicates preferential binding to the folded form compared with the unfolded form (Mergny JL, Lacroix L., Oligonucleotides, 2003;13(6):515-37). In experimental it is stated that every DTm value here reported was the average of at least two measurements. As for some weak stabilization effects (Figure 5, Table 3), we confirmed stabilization effects noticed for 1, 5 and 6 ligands and poly rA-poly dT with an RNase H experiment where ligands 1 and 5 were shown as better inhibitors than ligand 6 which showed smaller DTm value than 1 and 5. Regarding most interesting result, 6 showed strong stabilization effect of ATT (~44oC, Table 4, Figure 7); when including an error of ±0.5oC it is still a significant number.

Comment 8. Figure S1: Some spectra have a significant break at ca. 360 nm. Please explain.

Answer: Yes, some spectra in Figure S1 and some other Figures in SI have a significant break at 350 nm but this „break“ is unfortunately due to the transition from visible to UV lamp on our UV/Vis spectrophotometer.

Comment 9. The choice of nucleic acids should be justified in more detail, especially as most of them are not really biologically relevant. Especially, after particular selectivities of ligands have been identified, more elaborate and distinct oligonucleotide sequences with the same DNA/RNA structure should have been used to substantiate the general statements about selectivity and affinity towards this nucleic acid form.

Answer: We studied interactions of these compounds with long synthetic polynucleotides, polynucleotide form natural origin, ctDNA, and synthetic polynucleotides, poly(dAdT)2, poly(dGdC)2, poly dA – poly dT, poly A – poly U, hybrid polynucleotides poly dA-poly rU and poly rA – poly dT and ATT triplex. There are several reasons why are long synthetic polynucleotides chosen for study instead of short oligonucleotides. First, short oligomers are often not suitable due to the “capping” binding of heterocyclic moieties at the outer side of alternating basepairs, which strongly compete with the few binding sites along oligo-double strand (Casagrande V, Alvino A, Bianco A, Ortaggi G, Franceschin M., J Mass Spectrom. 2009 Apr;44(4):530-40.). Further, at variance to oligomers, polynucleotides consisting of >100 basepairs can assure large excess of binding sites along double helix, whereby the “capping” effect can be neglected. In addition, double stranded DNA and RNA are in biologically relevant conditions mostly long polynucleotides therefore for instance short oligomers can fail in representing a biologically significant structural model. Nevertheless, we repeated the most interesting result, complex of benzothiazole 6 with ATT polynucleotide, on shorter sequence, ATT oligonucleotide, 26mer (3.2.4. Circular dichroism (CD) experiments and Figures S23 and S35). The binding constant and ICD profile obtained for this shorter ATT sequence agreed very well with those ones obtained for the longer ATT sequence.

Comment 10. Figure 1: Although this figure is well organized and very informative, it is not necessary in this context and would rather serve its purpose in a review article.

 Answer: Thank you for your suggestion. Nevertheless, our opinion is that based on this image, the reader can more easily gain insight into the relevance of DNA / RNA hybrids and triplexes.

Comment 11. The literature on triplex ligands is not complete; and the list of refs (2–7) also contains literature on G4 DNA ligands, that is incomplete as well.

Answer: We listed, to our opinion, several relevant references on triplex ligands and less on G4 DNA ligands, since we didn't perform experiments with G-quadruplexes in this paper. Except for references 2-7, other references in list refer to triplex ligands (ref. 43 and 44 in previous version, 54 and 55 in revised version, ref. 62, 63 and 64 in previous version, 73, 74 and 75 in revised version). Nevertheless, according to reviewer suggestion, we inserted more relevant references on triplex ligands in Introduction and Conclusion (ref. 12, 13, 14 and 15 in Introduction and 76, 77 in Conclusion).

Comment 12. It is very disturbing that Figures and Tables proceed over page breaks in the Supporting Information.

Answer: This has been corrected in SI.

Comment 13. The conclusion is only a summary.

Answer: Thank you for your remark, we agree with you. We rewrote the conclusion, hopefully now is more concise and better.

Reviewer 2 Report

Recognition of ATT triplex and DNA:RNA hybrid structures by benzothiazole ligands

In this work, the interactions of an array of nucleic acid structures with a small series of benzothiazole ligands (bis-benzothiazolyl-pyridines – group 1, 2-thienyl/2-benzothienyl-substituted 6-(2-imidazoli-nyl)benzothiazoles – group 2, and three 2-aryl/heteroaryl-substituted 6-(2-imidazolinyl)benzothiazoles – group 3) were screened by competition dialysis.

The work presented by the authors is of very good quality, they determined the greatest number of parameters to determine the interaction of benzothiazoles with peptides, including the kinetics and the determined values.

However, in the methodology it does not mention the considerations used for molecular dynamics, as well as the data used to perform it, such as the center and the box, in addition to the time used in the calculation, the program and many data necessary in molecular dynamics.

The measurements made at work do not have error or standard deviation bars, it is necessary to include this information. Condense in the supplementary material a table with the triplicate values, means and deviation.

Author Response

Comment 1. However, in the methodology it does not mention the considerations used for molecular dynamics, as well as the data used to perform it, such as the center and the box, in addition to the time used in the calculation, the program and many data necessary in molecular dynamics.

Answer: We thank the reviewer for noticing that we did not put a methodological part regarding molecular dynamics. We inserted that in section 2: Materials and methods.

Comment 2. The measurements made at work do not have error or standard deviation bars, it is necessary to include this information. Condense in the supplementary material a table with the triplicate values, means and deviation.

Answer: In Scatchard equation values of stability constant (Ks) and ratio (n=[bound compound] / [polynucleotide]) are highly mutually dependent and similar quality of fitting calculated to experimental data is obtained for ±20% variation for Ks and n; this variation can be considered as an estimation of the errors for the given binding constants. Given estimation was added to Footnote of Table 2. Furthermore, this type of experiments are not usually performed in multiplicate, the reproducibility is confirmed by repeating some of the titrations (n=2), therefore deviations could not be calculated. Standard deviations of the fitting model for ITC are added in the supplementary material (Figure S24 and Figure S25.

Large errors with Scatchard analysis are often encountered (please see I. R. Klotz, Ligand-Receptor Energetics, John Wiley & Sons, Inc. New York, 1997). Since the concentration of observable species can determine the number of binding sites reflected in the isotherm, the apparent stoichiometry can change based upon the concentration of the observable species. In addition, the errors associated with assigning spectral properties of the 100% “free“ versus the 100% “bound” become amplified in all the data points, since the fraction bound at each data point is calculated from these two extremes. The data points for the 100% free and the 100% bound states are, therefore, “weighed” much more heavily than the points in the middle of the titration.

On the other hand, non-linear analysis of binding data can help reduce the errors associated with quantifying the spectral properties of these “extreme” (and often inaccurate) data points. Non-linear analysis typically weighs all data points equally and fits all the points to a theoretical curve. However, it is advisable to carefully choose experimental conditions to assure that all dye molecules bind to dominant binding sites – this is done by preliminary experiment for rough estimation of binding affinity and then repeating more detailed titration at conditions of an excess of DNA/RNA binding sites over c(dye), which allows each dye molecule to find its dominant binding site according to J.D. Mc Ghee, P.H. von Hippel formalism for non-cooperative binding (J. D. McGhee and P. H. von Hippel, J. Mol. Biol., 1976, 103, 679.,ref. 50 in the manuscript). More detailed considerations how to organize titration experiment and analysis are nicely summarised in J. Lah and G. Vesnaver, J Mol Biol, 2004, 342, 73 (pp 80).

As mentioned in Experimental, every thermal melting curve in thermal melting experiments is done in duplicate due to possible fluctuations in melting curves resulting for instance from entrapped air bubbles or slight precipitation of ligand – polynucleotide complex.

This data is inserted in the appropriate section in SI and marked in yellow.

Round 2

Reviewer 2 Report

All requested changes were made by the authors. The work presented has improved a lot in its quality. This can be accepted in this form.

Author Response

Thank you for the nice comment.